# Biphasic Calcium Phosphate Sphere Graft Combined with a Double-Layer Non-Crosslinked Collagen Membrane Technique for Ridge Preservation: A Randomized Controlled Animal Study

**DOI:** 10.3390/ma13010018

**Published:** 2019-12-18

**Authors:** Jungwon Lee, Young-Jun Lim, Bongju Kim, Ki-Tae Koo, Yong-Moo Lee

**Affiliations:** 1Department of Periodontology, One-Stop Specialty Center, Seoul National University Dental Hospital, Seoul 03080, Korea; jungwonlee.snudh@gmail.com; 2Department of Prosthodontics and Dental Research Institute, School of Dentistry, Seoul National University, Seoul 03080, Korea; 3Dental Life Science Research Institute & Clinical Translational Research Center for Dental Science, Seoul National University Dental Hospital, Seoul 03080, Korea; 4Department of Periodontology and Dental Research Institute, School of Dentistry, Seoul National University, Seoul 03080, Korea; periokoo@snu.ac.kr (K.-T.K.); ymlee@snu.ac.kr (Y.-M.L.)

**Keywords:** membrane, collagen, bone regeneration

## Abstract

The purpose of this study was to compare the histologic and radiologic differences between single- and double-layer collagen membrane techniques in flapless ridge preservation. The mandibular fourth premolar and first molar of four beagle dogs were used in the experiment. Mesial roots of the teeth were extracted and root canal treatment was performed at the distal roots. Ridge preservation was performed at the extraction sites using synthetic bone graft material. A single layer (control group) or double layer (test group) of non-crosslinked collagen membrane was applied following bone graft application. Three months later, the animals were sacrificed and micro-computed tomography (micro-CT) and histomorphometric analyses were conducted. Nonparametric Mann–Whitney test was performed to compare between the control and test groups. The vertical difference between buccal and lingual crests of control and test groups was 1.28 ± 0.41 and 0.53 ± 0.37 mm, respectively (*p* = 0.026). The mineralized bone area in control and test groups was 31.48% ± 7.41% and 42.25% ± 9.73%, respectively (*p* = 0.041). Within the limit of this study, ridge preservation using the double-layer membrane technique showed a reduced buccal bone resorption and improved new-bone formation in the ridge compared to that using the single-layer membrane technique.

## 1. Introduction

Following tooth extraction, rapid alveolar bone reduction occurs. Alveolar bone resorption can be observed within three months from tooth extraction [1], showing an alveolar contraction of 0.8 mm vertically and 50% horizontally [2]. These results are in agreement with a recent systematic review demonstrating alveolar ridge alteration after a non-infected extraction [3].

To prevent alveolar ridge contraction, alveolar preservation has been suggested [4,5,6,7,8]. However, alveolar preservation methods, including flap management, bone graft material, membrane, and suture procedure, have not yet been established very well, and research is ongoing to improve the effectiveness of ridge preservation. According to a recent systematic review, there was no significant difference in the amount of hard tissue altered between the primary closure group and secondary healing group, showing an increased keratinized gingiva width in secondary healing group compared to that in primary closure group [9]. Therefore, flapless ridge preservation, inducing secondary intention healing, may provide the expected clinical results, with less pain and discomfort, in case of a four-wall extraction socket.

On the other hand, while performing flapless ridge preservation using secondary intention healing, attachment of oral bacteria on the exposed membrane is inevitable [10]. Oral bacteria can produce proteinases that dissolve collagen [11]. As a result, degradation of the membrane proceeds more rapidly than in the case of primary closure [12]. If there would be a material or technique that could further improve the degradation resistance of the membrane, it might possibly reduce the exfoliation of bone graft during soft tissue healing and provide more effective alveolar preservation.

Biodegradation of non-crosslinked collagen membrane is known to require approximately 2–4 weeks [13]. Soft tissue wound healing in the oral cavity takes approximately 4–6 weeks [14]. Therefore, it is necessary to find alternative ways to increase resistance of the membrane or improve soft tissue wound healing around the extraction socket so that the collagen membrane is retained.

A recent study showed that the collagen membrane in flapless ridge preservation could prevent the escape of bone graft materials [15]. Previous study has demonstrated the application of double-layer non-crosslinked collagen membrane to increase resistance of the latter to biodegradation [16]. Another study had reported the reduction of membrane biodegradation using a double layer technique to improve the outcome of guided bone regeneration [17]. However, since these data were obtained without the membrane being exposed, it was considered necessary to investigate whether the results of ridge preservation could be improved using double-layer technique in flapless ridge preservation.

The purpose of this study was to compare the histologic and radiologic differences between single- and double-layer collagen membrane techniques in flapless ridge preservation.

## 2. Materials and Methods

### 2.1. Animal

The current study was approved by the Institutional Animal Care and Use Committee of Seoul National University (IACUC, SNU-170123-1-1). Four beagle dogs (Orient Bio, Seoul, Korea), weighing approximately 10 kg (age: 8 to 9 months), were used in this study. The animals were individually housed in 9 m in width, 8 m in depth, and 8 m in height indoor, ambient temperature 23 ± 2 °C, and relative humidity 50% ± 10%. Standard pellet food (HappyRang, Seoulfeed company, Seoul, Korea) was given throughout the present study and provided free access to water. The experimental time-table is shown in Figure 1. The study was conducted according to the ARRIVE guidelines [18].

### 2.2. Ridge Preservation

Research manuscripts reporting large datasets that are deposited in a publicly available database should specify where the data have been deposited and provide the relevant accession numbers. If the accession numbers have not yet been obtained at the time of submission, please state that they will be provided during review. They must be provided prior to publication of general anesthesia was performed by intravenous injection of Zoletil (0.1 mg/kg; Virbac, Carros, France), Rompun (2.3 mg/kg; Bayer Korea, Ansan, Korea), and atropine sulfate (0.05 mg/kg; Jeil, Daegu, Korea). In addition, local anesthesia was performed by injecting 1:100,000 epinephrine-containing lidocaine (Huons Co. Ltd., Seongnam, Korea). Periapical radiograph was taken before surgery. The mandibular fourth premolar (P4) and first molar (M1) were hemisected using a diamond bur, and the mesial root was extracted. Root canal treatment was performed on the remaining distal root, and the cavity was filled with intermediate restorative material (IRM, Dentsply, York, PA, DE, USA) (Figure 2). Alveolar ridge preservation was performed using a synthetic bone graft (Biphasic calcium phosphate sphere; HansBiomed, Seoul, Korea). A resorbable non-crosslinked collagen membrane (Bio-Gide; Geistlich, Wolhusen, Switzerland) was applied randomly using a single-layer (control group) or double-layer (test group) technique. Thereafter, suture was performed with 4/0 vicryl, and periapical radiograph was taken (Figure 3).

### 2.3. Post-Operative Protocol

In order to avoid post-operative trauma, a soft diet was provided. After alveolar ridge preservation, antibiotics and nonsteroidal anti-inflammatory drugs were administered. One week later, sutures were removed and plaque control was performed twice a month. Periapical X-rays were taken at 2, 4, 8, and 12 weeks post-operatively to identify any adverse reaction at the experimental sites.

### 2.4. Sacrifice and Block Biopsy

Twelve weeks after the surgery, the animals were sacrificed by injection of potassium chloride through the carotid artery, and the experimental sites were harvested. The extracted tissue was fixed in formalin for a week and demineralized in 5% formic acid for 10 days.

### 2.5. Micro-CT Analysis

After fixing the tissue block with the fixative solution, micro-CT images of the alveolar preservation site were obtained using a micro-CT device (SkyScan-1173, Kontich, Belgium). A two-dimensional image of 2240 × 2240 pixels was saved as a BMP file. Radiation exposure conditions while obtaining a micro-CT image were as follows: Voltage 92 kV, current 120 mA, exposure time 500 ms, aluminum filter of 1.0 mm, and image pixel size 19.18 μm. Projection image was reconstructed to generate 3D images, and analyzed using a CTAn software (Bruker, Kontich, Belgium).

Bone morphometric parameters for each volume of interest (VOI) were measured using the micro-CT images obtained. VOI in this study was the mesial root region of P4 and M1, where alveolar ridge preservation was performed. Bone morphometric parameters were named according to standard terms of the American Society for Bone and Mineral Research Histomorphometry Nomenclature Committee [19].

Vertical distance between the buccal and lingual crests was determined as per our previous study [17]. Briefly, a central vertical line (CVL) was set at the center of the extraction socket, and a horizontal line perpendicular to CVL was drawn from it to meet the buccal crest (BC) and lingual crest (LC), separately. Vertical distance was measured as the distance between two horizontal lines. Three horizontal lines, 1, 2, and 4 mm below the lingual crest, were drawn perpendicular to the CVL. The horizontal width of alveolar ridge at 1, 2, and 4 mm below lingual alveolar crest was defined as the horizontal distance between the buccal and lingual bones (Bw1, Bw2, and Bw4, respectively).

### 2.6. Histologic Processing

The biopsy-block was cut into 3 μm thick sections parallel to the long axis of extraction socket, and the two sections of middle-most area were stained with hematoxylin and eosin and Masson’s trichrome, respectively. Histological analysis was performed using an optical microscope (DP72; Olympus, Tokyo, Japan) and an imaging system (DP Controller; Olympus).

### 2.7. Histologic and Histomorphometric Analysis

Any inflammatory sign and adverse reaction in the process of healing of soft and hard tissues in each specimen were evaluated next. Histological analysis was performed using an optical microscope (DP72; Olympus, Tokyo, Japan) and an imaging system (DP Controller; Olympus). The scanned image of each specimen was saved as a 2400 × 1271 pixel JPG file, and an imaginary line from extraction socket was drawn on the area where the bone graft material was applied to set the region of interest. The ratio of connective tissue area, graft material area, and mineralized bone area was determined using an image measurement program (TOMORO ScopeEye version3.6.6 SARAMSOFT CO., Ltd. Anyang-si, Gyeonggi-do, Korea).

### 2.8. Statistical Analysis

Since this experiment was a pilot study, no sample size was obtained. Post hoc power analysis in vertical distance using G*Power version 3.1 software (Heinrich Heine Universität Düsseldorf, Düsseldorf, Germany; www.gpower.hhu.de) revealed that the observed statistical power was 94.59% with an alpha level of 0.05. Statistical analysis was performed using the SPSS version 17 program (IBM Software, Armonk, NY, USA). Radiological measurements and histomorphometric results are presented as mean ± standard deviation. Difference between the two groups was analyzed using the Mann–Whitney test.

## 3. Results

### 3.1. Clinical Findings

Soft tissue healing was appreciable at all alveolar ridge preservation sites. Inflammation was not observed in either of the two groups, treated with a single-layer (control) or double-layer membrane (test).

### 3.2. Micro-CT Analysis

Vertical distance between the buccal and lingual crests and ridge width at 1, 2, and 4 mm from the lingual alveolar crest (Bw1, Bw2, and Bw4, respectively) using 2D data, and bone morphometric analysis using 3D data, are shown in Table 1. Vertical distance between the buccal and lingual crests was 1.28 ± 0.41 mm in the control group and 0.53 ± 0.37 mm in the test group, which was statistically significant (*p* = 0.028).

Bw1, Bw2, and Bw4 were 4.46 ± 0.73, 5.88 ± 1.10, and 6.89 ± 0.81 mm in the control group and 4.72 ± 0.83, 5.22 ± 1.35, and 6.25 ± 1.43 mm in the test group, respectively. There was no statistically significant difference between the two groups with respect to radiographic bone width.

The BV/TV value was 56.41% ± 9.91% in the control group and 57.00% ± 10.13% in the test group. BS/TV was 10.69 ± 1.35 in the control group and 10.21 ± 1.98 in the test group. TbPf was 1.14 ± 1.07 in the control group and 0.49 ± 1.51 in the test group. SMI was 0.60 ± 0.54 in the control group and 0.25 ± 0.80 in the test group. There was no statistical difference between the two groups as per three-dimensional micro-CT analyses.

### 3.3. Hitologic Observation

No adverse reaction was observed upon bone healing, although cortical bone formation did not complete in most slides, thus implying incomplete bone healing. Cortical bone was not formed in the buccal bone of the extraction socket, as well as in the crestal area of extraction socket. It was not completely formed in either control or test group; however, cortical bone in the crestal and buccal portions of extraction socket was relatively well formed in the test group compared to that in the control group (Figure 4, Figure 5, Figure 6 and Figure 7).

Woven bone formation was observed in the apical portion of extraction socket and de-novo bone formation was observed on the surface of the graft material. Multinucleate giant cell was observed around the bone graft material, although no foreign body reaction was observed (Figure 5d and Figure 7d). Distribution of blood vessels was actively formed around the bone graft where new bone formation was actively processed, and boundary of the bone graft material itself was obscure, partially due to absorption of the bone graft. The bone graft material near the exterior of extraction socket was seen to be surrounded by multinucleated giant cells; however, the distribution of blood vessels and new bone formation was relatively less compared to that in the interior of extraction socket area (Figure 5d,e and Figure 7d,e).

### 3.4. Histomorphometric Analysis

In control group, 22.37% ± 12.18% of connective tissue, 46.15% ± 10.99% of bone graft materials, and 31.48% ± 7.41% of mineralized tissue were observed. In the test group, however, 19.88% ± 9.45% of connective tissue, 37.87% ± 11.58% of bone graft materials, and 42.25% ± 9.73% of mineralized tissue were observed. There was no statistical difference in connective tissue and bone graft material between the two groups; nevertheless, there was a significant difference in mineralized bone content between the two groups (Table 2).

## 4. Discussion

Alveolar bone contraction following tooth extraction has been demonstrated in various studies [1,2,3]. To counteract this ridge resorption, biphasic bone graft materials has been investigated for ridge preservation [20,21]. The positive effect of non-crosslinked collagen membrane in ridge preservation was also demonstrated in the previous study [15]. However, there is little evidence in flapless ridge preservation using biphasic bone graft materials with a double-layer technique.

In this experiment, we examined the differences in flapless ridge preservation between using single- and double-layer membrane techniques. Application of the latter in flapless ridge preservation, using biphasic bone graft materials, improved new bone formation and reduced the vertical distance between the buccal and lingual crests compared to that of single-layer membrane technique.

When performing the experiment using micro-CT images, vertical distance between the buccal and lingual crests showed a statistically significant difference between the control and test groups. This difference is attributed to the applied double-layer membrane not being absorbed, and providing support until the soft tissue healed and hard tissue formed.

Bone volume density (BV/TV), BS/BV, TbPf, and SMI were not affected by the number of membrane layers applied. It might be due to their representation of the inside of the extraction socket. However, the number of membranes applied on top of the extraction socket seems to affect the boundary of extraction socket. Therefore, there was no statistically significant difference in BV/TV, BS/BV, TbPf, and SMI. In Bw1, Bw2, and Bw4 as well, there was no difference between the two groups. Given that the vertical distance between buccal and lingual crests was approximately 1 mm in both groups, the bone graft inside the extraction socket, placed 1 mm from the lingual crest, was almost stable. These results are similar to those of previous studies that investigated volumetric changes [22].

In histomorphometric analysis, a higher percentage of mineralized bone was observed in the test group compared to that in the control group. This could be because the ridge preserved using double-layer membrane technique possibly functions as a barrier membrane when new bone formation is processed inside. Our previous study had demonstrated GBR procedures, using double-layer membrane technique, to retain the membrane up to six months, which reduced bone graft material resorption more than in the single-layer membrane application [17]. Unlike the previous study, present study was performed with the membrane exposed. Thickness of the non-crosslinked collagen membrane rapidly decreases, since biodegradation occurs two weeks after exposure of the membrane to the oral cavity [13]. Active vascularization was observed inside the membrane, since the membrane itself was absorbed [13]. Due to this biocompatibility, soft tissues in the upper part of the extraction socket healed well, thereby preventing the exfoliation of bone graft material. On the other hand, when the upper membrane was absorbed and healed into soft tissues, the lower membrane was retained for a certain period of time, and could play a positive role in the formation of new bone within the extraction socket.

Although quantitative measurement cannot be performed, it is more likely to have formed cortical bone in the test group than in the control group, when histologic examination was performed. Previous study had shown a positive effect on ridge preservation with non-crosslinked membrane compared to that in the no-membrane group [17]. The positive role of non-crosslinked membrane in ridge preservation was further improved by the double-layer membrane technique, by prolonging the retention period of the membrane.

Clinically, tooth extraction is often required due to periodontitis and combined endodontic–periodontic lesions [23]. In this regard, there is a limitation in this study, since ridge preservation was performed on an intact extraction socket. Moreover, another limitation was regarding soft tissue healing and membrane resistance, which was not investigated when single- or double-layer membrane was applied. Therefore, ridge preservation with a double-layer membrane technique in infected extraction socket should be confirmed in terms of soft tissue and hard tissue healing in future/upcoming studies.

## 5. Conclusions

Within the limit of this study, ridge preservation using the double-layer membrane technique showed a reduced buccal bone resorption and improved new-bone formation in the ridge compared to that using the single-layer membrane technique.

## Figures and Tables

**Figure 1 materials-13-00018-f001:**
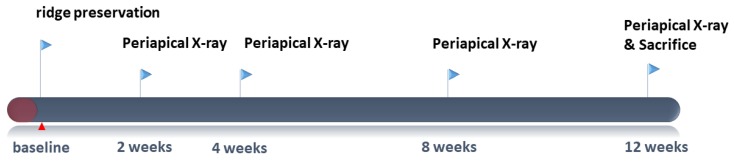
Timeline of the experiment.

**Figure 2 materials-13-00018-f002:**
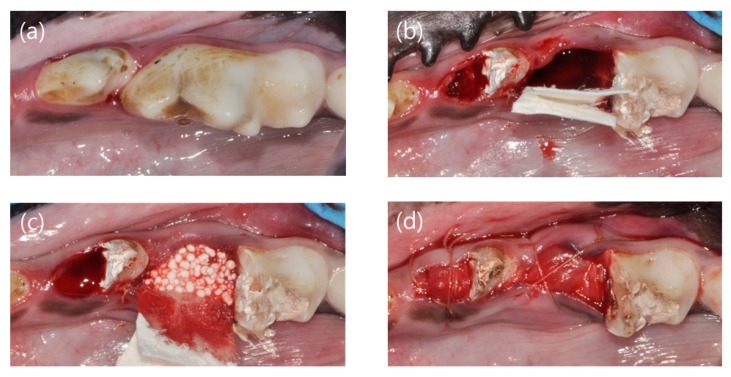
Clinical photographs showing the experimental procedures. (**a**) Before surgery, (**b**) mesial root extraction and membrane application, (**c**) bone graft material application, and (**d**) suture.

**Figure 3 materials-13-00018-f003:**
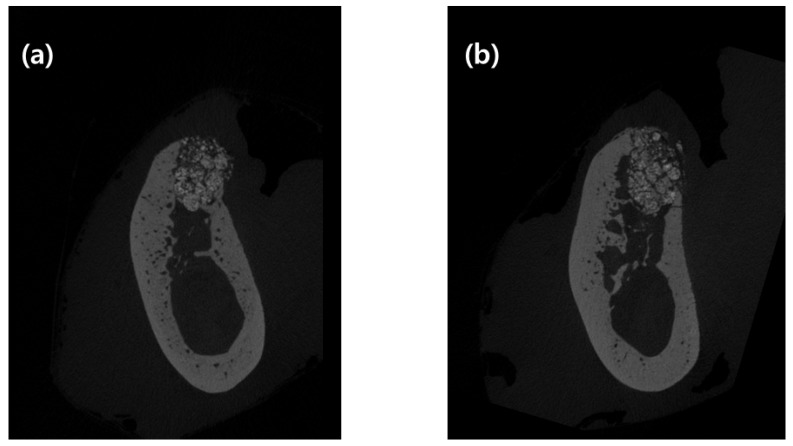
Buccolingual micro-computed tomography (micro-CT) images of two groups of ridge preservation. (**a**) Single-layer membrane application and (**b**) double-layer membrane application. CT images were selected as 3 mm mesial from the experimental septum of tooth.

**Figure 4 materials-13-00018-f004:**
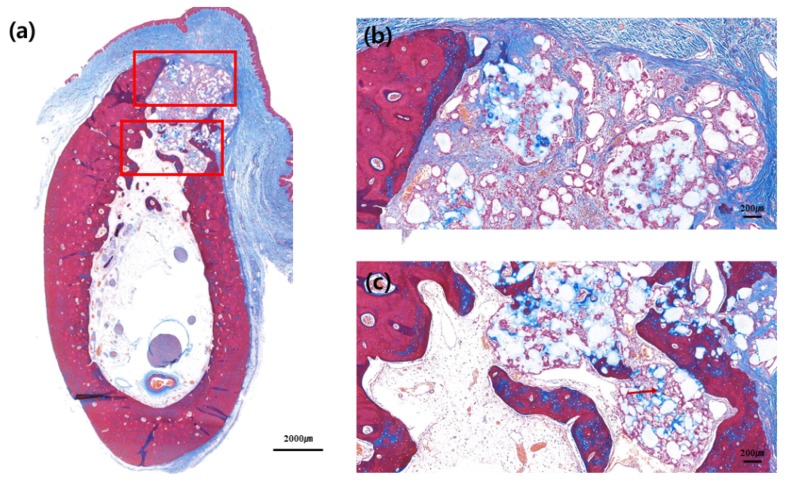
Buccolingual section at three months after ridge preservation in the control group. Masson trichrome staining: (**a**) Alveolar bone at the experimental site, (**b**) crestal area, and (**c**) apical area. Red arrow indicates the new bone formed around bone graft materials.

**Figure 5 materials-13-00018-f005:**
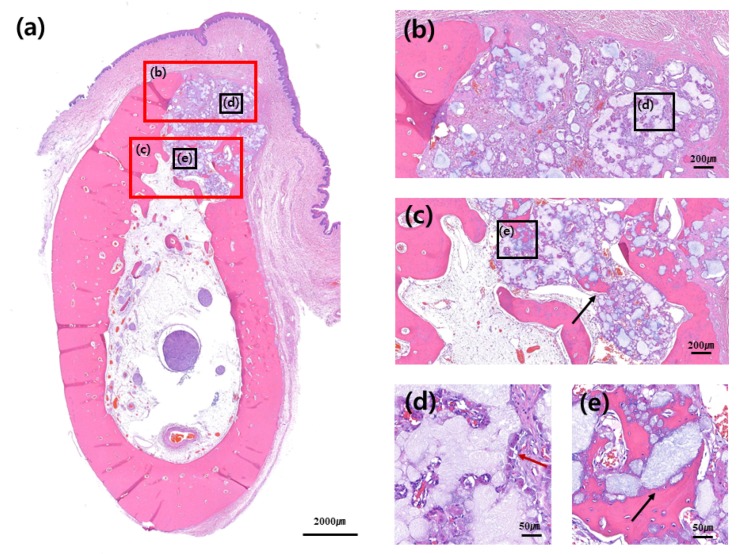
Buccolingual section at three months after ridge preservation in the control group. Hematoxylin and eosin staining: (**a**) Alveolar bone at the experimental site, (**b**) crestal area, (**c**) apical area, (**d**) buccal and crestal area, and (**e**) and inner area. Black arrow indicates de novo bone formation around the bone graft materials. Red arrow indicates multinucleated giant cells around the bone graft materials.

**Figure 6 materials-13-00018-f006:**
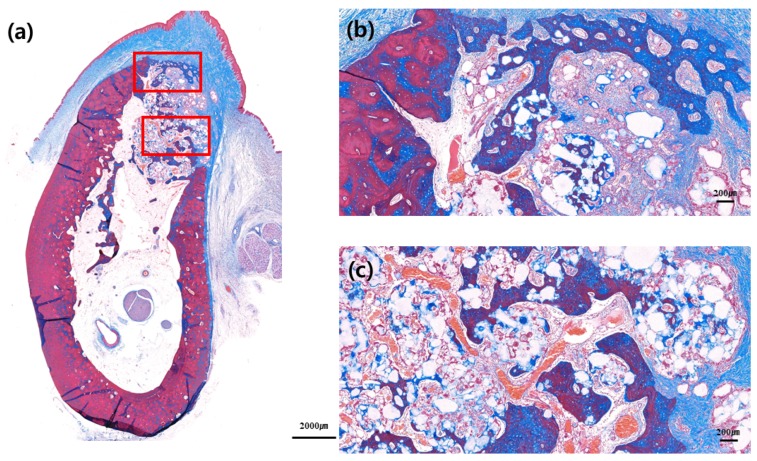
Buccolingual section at three months after ridge preservation in the test group. Masson trichrome staining: (**a**) Alveolar bone at the experimental site, (**b**) crestal area, and (**c**) apical area. Compared to the control group, the hard tissue bridge in the upper part of the buccal crest is more noticeable. Red arrow indicates new bone formation around bone graft materials.

**Figure 7 materials-13-00018-f007:**
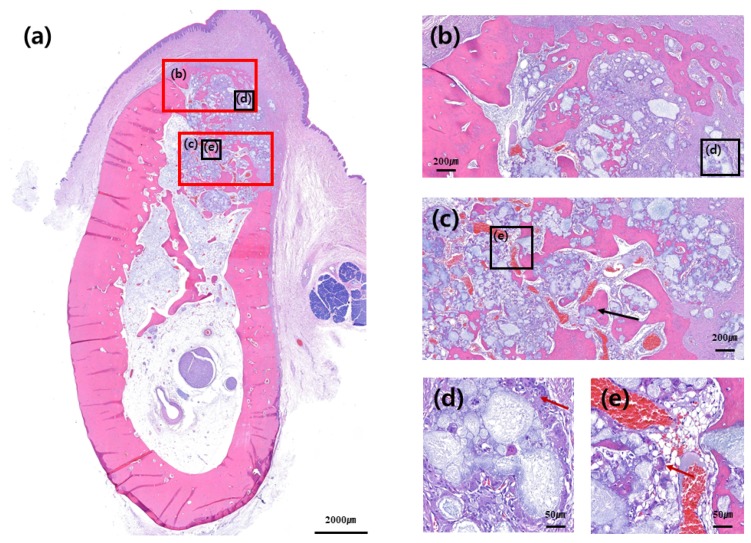
Buccolingual section at three months after ridge preservation in the test group. Hematoxylin and eosin staining: (**a**) Alveolar bone at the experimental site, (**b**) crestal area, (**c**) apical area, (**d**) buccal and crestal area, and (**e**) apical and inner area. Even if multinucleated giant cells were observed around the bone graft, new bone formation was more visible inside the extraction socket, where a high distribution of blood vessels was observed. Black arrow indicates de-novo bone formation around the bone graft materials. Red arrow indicates multinucleated giant cells around the bone graft materials.

**Table 1 materials-13-00018-t001:** Micro-CT two-dimensional (2D) and three-dimensional (3D) analyses of control and test group (mean ± standard deviation).

Groups	N	Vertical Distance (mm)	Bw1 (mm)	Bw2 (mm)	Bw4 (mm)	BV/TV (%)	BS/TV	TbPf	SMI
Control	8	1.28 ± 0.41	4.46 ± 0.73	5.88 ± 1.10	6.89 ± 0.81	56.41 ± 9.91	10.69 ± 1.35	1.14 ± 1.07	0.60 ± 0.54
Test	8	0.53 ± 0.37	4.72 ± 0.83	5.22 ± 1.35	6.25 ± 1.43	57.00 ± 10.13	10.21 ± 1.98	0.49 ± 1.51	0.25 ± 0.80
*p*-value		0.026	0.699	0.310	0.310	1.000	0.907	0.381	0.569

The vaules of the parameters were measured with an accuracy of 0.01. Bw1, Bw2, and Bw4 indicate the bucco-lingual bone width at 1, 2, and 4 mm below the lingual crest. *p*-value means a nonparametric Mann–Whitney test.

**Table 2 materials-13-00018-t002:** Results of histomorphometric analysis of control and test group (mean ± standard deviation).

Groups	N	Connective Tissue (%)	Bone Graft Materials (%)	Mineralized Bone (%)
Control	8	22.37 ± 12.18	46.15 ± 10.99	31.48 ± 7.41
Test	8	19.88 ± 9.45	37.87 ± 11.58	42.25 ± 9.73
*p*-value		0.818	0.589	0.041

The vaules of the parameters were measured with an accuracy of 0.01. *p*-value means a nonparametric Mann–Whitney test.

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
