# Peer review of "Biphasic Calcium Phosphate Sphere Graft Combined with a Double-Layer Non-Crosslinked Collagen Membrane Technique for Ridge Preservation: A Randomized Controlled Animal Study"

_materials, 2019, doi:10.3390/ma13010018_

Round 1

Reviewer 1 Report

This manuscript untitled “Biphasic Calcium Phosphate Sphere Graft Combined 2 with a Double-Layer Non-Crosslinked Collagen 3 Membrane Technique for Ridge Preservation: an 4 Experimental Study in Dogs” is a well-conducted study, good presentation, easy to read. Aim of this paper is quite interesting, but is not new theme. Generally, there are some grammatical errors in this manuscript. May be, it is recommended that it would be revised again by English scientific writer.

The introduction is direct to the objective that you want to study, the research is well done. In Methodology the statistical analyze was adequate. This evaluation method is objective. Aim of the study did correspond its conclusion. Results of the study induce this conclusion. The references are standardized and well inserted in the manuscript.

However, I had some concerns about this work:

I suggest change the title for “: experimental animal study” (Study Design) How did you perform the sample calculation? When mentioning materials or devices: you don't mention the manufacturer, city and the country(HappyRang, Seoulfeed company, City; Korea). It would be better if you make the same mentions each time you refer to a commercial product.

Author Response

Dear Reviewer 1

This manuscript untitled “Biphasic Calcium Phosphate Sphere Graft Combined 2 with a Double-Layer Non-Crosslinked Collagen 3 Membrane Technique for Ridge Preservation: an 4 Experimental Study in Dogs” is a well-conducted study, good presentation, easy to read. Aim of this paper is quite interesting, but is not new theme. Generally, there are some grammatical errors in this manuscript. May be, it is recommended that it would be revised again by English scientific writer.

The introduction is direct to the objective that you want to study, the research is well done. In Methodology the statistical analyze was adequate. This evaluation method is objective. Aim of the study did correspond its conclusion. Results of the study induce this conclusion. The references are standardized and well inserted in the manuscript.

However, I had some concerns about this work:

I suggest change the title for “: experimental animal study” (Study Design)

We corrected the title according to your suggestion as follow:

Biphasic calcium phosphate sphere graft combined with a double-layer non-crosslinked collagen membrane technique for ridge preservation: an experimental study in dogs

Biphasic calcium phosphate sphere graft combined with a double-layer non-crosslinked collagen membrane technique for ridge preservation: a randomized controlled animal study

How did you perform the sample calculation?

The sample size calculation was not performed due to the nature of pilot study. We added a sentence related with statistical power in the paragraph.

Since this experiment was a pilot study, no sample size calculation was performed. Post hoc power analysis in vertical distance using G*Power version 3.1 software (Heinrich Heine Universität Düsseldorf, Düsseldorf, Germany; www.gpower.hhu.de) revealed that the observed statistical power was 94.59% with an alpha level of 0.05.

When mentioning materials or devices: you don't mention the manufacturer, city and the country(HappyRang, Seoulfeed company, City; Korea). It would be better if you make the same mentions each time you refer to a commercial product.

We corrected the sentence according to your suggestion as follow:

Standard pellet food (HappyRang, Seoulfeed company, Korea) was given thoughout the present study and provided free access to water.

Standard pellet food (HappyRang, Seoulfeed company, Seoul, Korea) was given thoughout the present study and provided free access to water.

Reviewer 2 Report

Materials and Methods. The text at lines 80-83 are instructions to the authors for this type of study and must be deleted.

Results. Brief comments are needed about the accuracy of the measurements for the data reported in Tables 1 and 2. Can the data be acceptably reported to the two decimal places shown for the values of means and standard deviations? Was the second decimal place generated by the statistical analysis?

Discussion. (1) Since this is a materials-oriented journal, there should be a new paragraph that concisely explains (with references) the composition and structure of the biphasic bone graft material and the collagen membrane material, noting their relationships to the structure of alveolar bone. (2) Another new paragraph at the end of this section should provide a summary of the limitations of this study and include suggestions for future research.

Author Response

Dear Reviewer 2

Materials and Methods. The text at lines 80-83 are instructions to the authors for this type of study and must be deleted.

The text was removed according to your suggestion as follow:

Research manuscripts reporting large datasets that are deposited in a publicly available database should specify where the data have been deposited and provide the relevant accession numbers. If the accession numbers have not yet been obtained at the time of submission, please state that they will be provided during review. They must be provided prior to publicationGeneral anesthesia was performed by intravenous injection of Zoletil (0.1 mg / kg; Virbac, Carros, France), Rompun (2.3 mg / kg; Bayer Korea, Ansan, Korea), and atropine sulfate (0.05 mg / kg; Jeil, Daegu, Korea). In addition, local anesthesia was performed by injecting 1: 100,000 epinephrine-containing lidocaine (Huons Co. Ltd., Seongnam, Korea).

General anesthesia was performed by intravenous injection of Zoletil (0.1 mg / kg; Virbac, Carros, France), Rompun (2.3 mg / kg; Bayer Korea, Ansan, Korea), and atropine sulfate (0.05 mg / kg; Jeil, Daegu, Korea). In addition, local anesthesia was performed by injecting 1: 100,000 epinephrine-containing lidocaine (Huons Co. Ltd., Seongnam, Korea).

Results. Brief comments are needed about the accuracy of the measurements for the data reported in Tables 1 and 2. Can the data be acceptably reported to the two decimal places shown for the values of means and standard deviations? Was the second decimal place generated by the statistical analysis?

According to your suggestion, we added the sentence as follow:

Table 1. Micro-CT 2D and 3D analyses of control and test group. (mean ± standard deviation).

Groups

N

Vertical distance (mm)

Bw1 (mm)

Bw2 (mm)

Bw4 (mm)

BV/TV (%)

BS/TV

TbPf

SMI

Control

8

1.28 ± 0.41

4.46 ± 0.73

5.88 ± 1.10

6.89 ± 0.81

56.41 ± 9.91

10.69 ± 1.35

1.14 ± 1.07

0.60 ± 0.54

Test

8

0.53 ± 0.37

4.72 ± 0.83

5.22 ± 1.35

6.25 ± 1.43

57.00 ± 10.13

10.21 ± 1.98

0.49 ± 1.51

0.25 ± 0.80

p-value

0.026

0.699

0.310

0.310

1.000

0.907

0.381

0.569

The vaules of the parameters were measured with an accuracy of 0.01.

Bw1, Bw2, and Bw4 indicate the bucco-lingual bone width at 1, 2, and 4 mm below the lingual crest.

P-value means a nonparametric Mann-Whitney test

Table 2. Results of histomorphometric analysis of control and test group (mean ± standard deviation).

Groups

N

Connective tissue (%)

Bone graft materials (%)

Mineralized bone (%)

Control

8

22.37 ± 12.18

46.15 ± 10.99

31.48 ± 7.41

Test

8

19.88 ± 9.45

37.87 ± 11.58

42.25 ± 9.73

p-value

0.818

0.589

0.041

The vaules of the parameters were measured with an accuracy of 0.01.
P-value means a nonparametric Mann-Whitney test.

Discussion.

(1) Since this is a materials-oriented journal, there should be a new paragraph that concisely explains (with references) the composition and structure of the biphasic bone graft material and the collagen membrane material, noting their relationships to the structure of alveolar bone.

According to your suggestion, we added one paragraph as follow:

Alveolar bone contraction following tooth extraction has been demonstrated in various studies [1,2,3]. To counteract this ridge resorption, biphasic bone graft materials has been investigated for ridge preservation [20,21]. The positive effect of non-crosslinked collagen membrane in ridge preservation was also demonstrated in the previous study [15]. However, there is little evidence in flapless ridge preservation using biphasic bone graft materials with double-layer technique.

(2) Another new paragraph at the end of this section should provide a summary of the limitations of this study and include suggestions for future research.

According to your suggestion, we added new paragraph related with the limitations of this study and suggestions for future study.

Clinically, tooth extraction is often required due to periodontitis and combined endodontic-periodontic lesions [21]. In this regard, there is a limitation in this study, since ridge preservation was performed on intact extraction socket. Moreover, another limitation was regarding soft tissue healing and membrane resistance, which was not investigated when single- or double-layer membrane was applied. Therefore, ridge preservation with double-layer membrane technique in infected extraction socket should be confirmed in terms of soft tissue and hard tissue healing in future/upcoming studies.

Reviewer 3 Report

This experimental investigation is a paper that presents information for clinicians in the field of regeneration with bone graft materials in dentistry. The aim of the study was to compare the histologic and radiologic differences between single- and double-layer collagen membrane techniques in flapless ridge preservation.

The Introduction not showed, today, the state of art. In the paragraph 4 of the second page, the authors showed :

Recent study has demonstrated the application of double-layer non-crosslinked collagen membrane to increase resistance of the latter to biodegradation [15]. Another study had reported the reduction of membrane biodegradation using double layer technique to improve the outcome of guided bone regeneration [16].

These studies are published 10 years ago.

Materials and methods. This section is showed correctly according scientific methodology. The authors defined the characteristics of the animals in relation to the surgery and regeneration of bone defects. Histologic assessment and tomography diagnosis are correctly included in this section.

NOTE. In the first paragraph  of the page 3:

A resorbable non-crosslinked collagen membrane (Bio-Gide; Geistlich, Wolhusen, Switzerland) was applied randomly using single-layer (control) or double-layer technique.

The text must be included in single-layer (control group) and (test group) after double-layer

Results. This section is showed correctly according scientific publications.  Micro-CT demonstrated that vertical difference between buccal and lingual crests of control and test groups was 1.28 ± 0.41 27 mm and 0.53 ± 0.37 mm, respectively.  Histologic study demonstrated no adverse reactions upon bone healing. Cortical bone was not formed in the buccal bone of the extraction socket as well as in the crestal area of extraction socket. It was not completely formed in either control or test group.   Histomorphometric study demonstrated that the mineralized bone area in control and test groups was 31.48 ± 7.41% and 42.25 ± 9.73%, respectively.

Discussion. Several paragraphs of this section repeat the results of the study, but is necessary a true discussion with more and recent observations published.

References are very older. Only 4 references (19%) are published at last 5 years.

Conclusively, the study is not accepted for publication and need major revision.

Author Response

Dear Reviewer 3

This experimental investigation is a paper that presents information for clinicians in the field of regeneration with bone graft materials in dentistry. The aim of the study was to compare the histologic and radiologic differences between single- and double-layer collagen membrane techniques in flapless ridge preservation.

The Introduction not showed, today, the state of art. In the paragraph 4 of the second page, the authors showed :

Recent study has demonstrated the application of double-layer non-crosslinked collagen membrane to increase resistance of the latter to biodegradation [15]. Another study had reported the reduction of membrane biodegradation using double layer technique to improve the outcome of guided bone regeneration [16].

These studies are published 10 years ago.

As your comment, we corrected the paragraph as follow:

Recent study showed the collagen membrane in flapless ridge preservation could prevent the escape of bone graft materials [15]. Previous study has demonstrated the application of double-layer non-crosslinked collagen membrane to increase resistance of the latter to biodegradation [16]. Another study had reported the reduction of membrane biodegradation using double layer technique to improve the outcome of guided bone regeneration [17]. However, since these data were obtained without the membrane being exposed, it was considered necessary to investigate whether the results of ridge preservation could be improved using double-layer technique in flapless ridge preservation.

Materials and methods. This section is showed correctly according scientific methodology. The authors defined the characteristics of the animals in relation to the surgery and regeneration of bone defects. Histologic assessment and tomography diagnosis are correctly included in this section.

NOTE. In the first paragraph of the page 3:

A resorbable non-crosslinked collagen membrane (Bio-Gide; Geistlich, Wolhusen, Switzerland) was applied randomly using single-layer (control) or double-layer technique.

 The text must be included in single-layer (control group) and (test group) after double-layer

According to your suggestion, we corrected the sentence as follow:
A resorbable non-crosslinked collagen membrane (Bio-Gide; Geistlich, Wolhusen, Switzerland) was applied randomly using single-layer (control) or double-layer technique.

 A resorbable non-crosslinked collagen membrane (Bio-Gide; Geistlich, Wolhusen, Switzerland) was applied randomly using single-layer (control group) or double-layer (test group) technique.

Results. This section is showed correctly according scientific publications.  Micro-CT demonstrated that vertical difference between buccal and lingual crests of control and test groups was 1.28 ± 0.41 27 mm and 0.53 ± 0.37 mm, respectively.  Histologic study demonstrated no adverse reactions upon bone healing. Cortical bone was not formed in the buccal bone of the extraction socket as well as in the crestal area of extraction socket. It was not completely formed in either control or test group.   Histomorphometric study demonstrated that the mineralized bone area in control and test groups was 31.48 ± 7.41% and 42.25 ± 9.73%, respectively.

Discussion. Several paragraphs of this section repeat the results of the study, but is necessary a true discussion with more and recent observations published.

According to your suggestion, we added a paragraph with recent published studies as follow:

Alveolar bone contraction following tooth extraction has been demonstrated in various studies [1,2,3]. To counteract this ridge resorption, biphasic bone graft materials has been investigated for ridge preservation [20,21]. The positive effect of non-crosslinked collagen membrane in ridge preservation was also demonstrated in the previous study [15]. However, there is little evidence in flapless ridge preservation using biphasic bone graft materials with double-layer technique.

References are very older. Only 4 references (19%) are published at last 5 years.

 We added recently published studies as follow:

Naenni, N.; Sapata, V.; Bienz, SP.; Leventis, M.; Jung, RE.; Hämmerle, CHF.; Thoma, DS.; Effect of flapless ridge preservation with two different alloplastic materials in sockets with buccal dehiscence defects—volumetric and linear changes. Clin Oral Invest 2018, 22: 2187. Ho KN.; Salamanca E.; Chang KC.; Shih TC.; Chang YC.; Huang HM.; Teng NC.; Lin CT.; Feng SW.; Chang WJ.; A Novel HA/β-TCP-Collagen Composite Enhanced New Bone Formation for Dental Extraction Socket Preservation in Beagle Dogs. Materials (Basel). 2016, 11;9(3).

Round 2

Reviewer 3 Report

The new version of the paper is correct.